# Effectiveness of Third-Class Biologic Treatment in Crohn’s Disease: A Multi-Center Retrospective Cohort Study

**DOI:** 10.3390/jcm10132914

**Published:** 2021-06-29

**Authors:** Ahmad Albshesh, Joshua Taylor, Edoardo V. Savarino, Marie Truyens, Alessandro Armuzzi, Davide G. Ribaldone, Ariella Bar-Gil Shitrit, Morine Fibelman, Pauliina Molander, Claire Liefferinckx, Stephane Nancey, Mohamed Korani, Mariann Rutka, Manuel Barreiro-de Acosta, Viktor Domislovic, Gerard Suris, Carl Eriksson, Catarina Alves, Afroditi Mpitouli, Caroline di Jiang, Katja Tepeš, Marina Coletta, Kalliopi Foteinogiannopoulou, Javier P. Gisbert, Hadar Amir-Barak, Mohamed Attauabi, Jakob Seidelin, Waqqas Afif, Carla Marinelli, Triana Lobaton, Daniela Pugliese, Nitsan Maharshak, Anneline Cremer, Jimmy K. Limdi, Tamás Molnár, Borja Otero-Alvarin, Zeljko Krznaric, Fernando Magro, Konstantinos Karmiris, Tim Raine, David Drobne, Ioannis Koutroubakis, Maria Chaparro, Henit Yanai, Johan Burisch, Uri Kopylov

**Affiliations:** 1Sheba Medical Center, Department of Gastroenterology, Sackler School of Medicine, Tel-Aviv University, Tel-Aviv 52621, Israel; ukopylov@gmail.com; 2Department of Gastroenterology, Montreal General Hospital, Montreal, QC 1650, Canada; joshua.taylor@mail.mcgill.ca (J.T.); waqqas.afif@mcgill.ca (W.A.); 3Division of Gastroenterology, Department of Surgery, Oncology and Gastroenterology, University of Padua, 35128 Padua, Italy; edoardosavarino@gmail.com (E.V.S.); carla.marinelli84@gmail.com (C.M.); 4IBD Unit, Department of Gastroenterology, Ghent University Hospital, 9000 Ghent, Belgium; Marie.Truyens@uzgent.be (M.T.); Triana.LobatonOrtega@uzgent.be (T.L.); 5IBD Unit, Fondazione Policlinico Universitario A. Gemelli IRCCS, Università Cattolica del Sacro Cuore, 00168 Rome, Italy; alearmuzzi@yahoo.com (A.A.); danielapugliese@gmail.com (D.P.); 6Division of Gastroenterology, Department of Medical Sciences, University of Turin, 10126 Turin, Italy; davrib_1998@yahoo.com; 7Shaare Zedek Medical Center, Faculty of Medicine, Digestive Diseases Institute, Hebrew University of Jerusalem, Jerusalem 9372212, Israel; ariellash@szmc.org.il; 8Tel Aviv Medical Center, Department of Gastroenterology and Liver Diseases, Sackler Faculty of Medicine, Tel-Aviv University, Tel-Aviv 6423906, Israel; morin.fei@gmail.com (M.F.); nitsanm@tlvmc.gov.il (N.M.); 9Abdominal Center, Department of Gastroenterology, Helsinki University Hospital, 00101 Helsinki, Finland; pauliina.molander@hus.fi; 10Department of Gastroenterology, Erasme University Hopital, 1070 Brussels, Belgium; claire2301@gmail.com (C.L.); anneline.cremer@erasme.ulb.ac.be (A.C.); 11Department of Gastroenterology, Hospices Civils de Lyon, University Claude Bernard Lyon, 69495 Lyon, France; stephane.nancey@chu-lyon.fr; 12INSERM, U1111, CIRI, 69007 Lyon, France; 13Division of Gastroenterology, The Pennine Acute Hospitals NHS Trust, Manchester M8 6RB, UK; Mohamed.Korani@pat.nhs.uk (M.K.); jimmy.limdi@nhs.net (J.K.L.); 14Manchester Academic Health Sciences, University of Manchester, Manchester BL97TD, UK; 15First Department of Medicine, University of Szeged, H-6720 Szeged, Hungary; mariannrutka@gmail.com (M.R.); molnar.tamas@med.u-szeged.hu (T.M.); 16IBD Unit, Gastroenterology Department, University Hospital of Santiago de Compostela, 15706 Santiago, Spain; manubarreiro@hotmail.com (M.B.-d.A.); borja.alvarin@gmail.com (B.O.-A.); 17Department of Gastroenterology, Hepatology, and Nutrition, University Hospital Centre Zagreb, 10000 Zagreb, Croatia; viktor.domislovic@gmail.com (V.D.); zeljko.krznaric1@zg.t-com.hr (Z.K.); 18Digestive System Service, Bellvitge University Hospital, Catalan Institute of Health, 08907 Barcelona, Spain; tsuris@bellvitgehospital.cat; 19Department of Gastroenterology, Faculty of Medicine and Health, Örebro University, SE-70182 Örebro, Sweden; carl.eriksson@regionorebrolan.se; 20Department of Gastroenterology, Centro Hospitalar São João, 4200319 Porto, Portugal; catarinalves_04@hotmail.com (C.A.); fernandomagro65@gmail.com (F.M.); 21Department of Gastroenterology, Venizeleion General Hospital, Heraklion, 71409 Crete, Greece; bitouliafro@hotmail.com (A.M.); kkarmiris@gmail.com (K.K.); 22Department of Gastroenterology, Addenbrooke’s Hospital, Cambridge University Hospitals, Cambridge CB2 0QQ, UK; caroline.di.jiang@gmail.com (C.d.J.); tim.raine@addenbrookes.nhs.U.K (T.R.); 23University Medical Centre Ljubljana, Department of Gastroenterology, Medical Faculty, University of Ljubljana, 1231 Ljubljana, Slovenia; tepes.katja@gmail.com (K.T.); david.drobne@gmail.com (D.D.); 24Department of Hepatology and Clinical Gastroenterology, ASST Santi Paolo e Carlo-Ospedale San Polo Universitario Milano Mariabeatrice, 20142 Milan, Italy; marina.coletta@gmail.com; 25Department of Gastroenterology, University Hospital of Heraklion, 71500 Crete, Greece; drfpopi@hotmail.com (K.F.); ikoutroubakis@gmail.com (I.K.); 26Centro de Investigación Biomédica en Red de Enfermedades Hepáticas y Digestivas (CIBEREHD), Hospital Universitario de La Princesa, Instituto de Investigación Sanitaria Princesa (IIS-IP), Universidad Autónoma de Madrid, 28001 Madrid, Spain; javier.p.gisbert@gmail.com (J.P.G.); mariachs2005@gmail.com (M.C.); 27IBD Center, Division of Gastroenterology, Rabin Medical Center, Beilinson Campus, Petah Tikva, Israel and the Sackler School of Medicine, Tel-Aviv University, Tel-Aviv 49100, Israel; Hadar.amir@gmail.com (H.A.-B.); henityanai@gmail.com (H.Y.); 28Copenhagen Center for Inflammatory Bowel Disease in Children, Adolescents and Adults, Hvidovre Hospital, University of Copenhagen, 2650 Copenhagen, Denmark; mohamed.attauabi.02@regionh.dk (M.A.); burisch@gmail.com (J.B.); 29Department of Gastroenterology, Herlev University Hospital, 3400 Copenhagen, Denmark; jakob.benedict.seidelin@regionh.dk

**Keywords:** Crohn’s disease, anti-TNF failure, treatment response, treatment failure, ustekinumab, vedolizumab

## Abstract

Background: Multiple studies have described the effectiveness of ustekinumab (UST) and vedolizumab (VDZ) in patients with Crohn’s disease (CD) failing anti- Tumor necrosis factors (TNFs); however, the effectiveness of VDZ or UST as a third-class biologic has not yet been described. Aims and Methods: In this retrospective multicenter cohort study, we aimed to investigate the effectiveness of VDZ and UST as a third-class biologic in patients with CD. Results: Two-hundred and four patients were included; 156/204 (76%) patients received VDZ as a second- and UST as a third-class therapy (group A); the remaining 48/204 (24%) patients received UST as a second- and VDZ as a third-class therapy (group B). At week 16–22, 87/156 (55.5%) patients and 27/48 (56.2%) in groups A and B, respectively, responded to treatment (*p* = 0.9); 41/156 (26.2%) and 15/48 (31.2%) were in clinical remission (*p* = 0.5). At week 52; 89/103 (86%) patients and 25/29 (86.2%) of the patients with available data had responded to third-class treatment in groups A and B, respectively (*p* = 0.9); 31/103 (30%) and 47/29 (24.1%) were in clinical remission (*p* = 0.5). Conclusion: Third-class biological therapy was effective in more than half of the patients with CD. No differences in effectiveness were detected between the use of VDZ and UST as a third-class agent.

## 1. Introduction

In recent years, our arsenal of therapeutic options in Crohn’s disease (CD) has been expanding. Despite the diversity of therapeutic options, loss of response remains a significant challenge across all therapeutic agents [1,2]. Tumor necrosis factor (TNF)-alpha inhibitors are the most frequently used first-class biologics; however, approximately 30% of patients experience primary and at least 13% experience secondary loss of response per year [3,4]. Both ustekinumab (UST) and vedolizumab (VDZ) are frequently used as second-class options, with response rates of 40–70% and loss of response rates approximating 20–30% within the first year [5,6,7,8,9,10,11,12,13,14,15,16,17,18,19,20,21,22,23,24,25,26]. As we are currently lacking knowledge regarding predictors of the response to either of these biologic classes of treatment [7], the decision is largely empirical.

Recently, two real-world studies compared the effectiveness of VDZ and UST in CD patients following loss of response to anti-TNFs [27,28]. In both studies, UST was superior to VDZ. However, loss of response to a subsequent second-class biologic is frequent. Moreover, the number of previously utilized biologics is associated with a diminished likelihood of response to biological therapy [29]. To date, there is scarce evidence regarding the effectiveness of a third-class biologic after the failure of two previous classes in CD [30].

The current study aimed to evaluate the likelihood and potential predictors of response to a third-class biologic in CD patients with a previous failure of two biologic classes.

## 2. Materials and Methods

### 2.1. Patient Population

This was a multicenter retrospective cohort study. We included adult patients with an established CD diagnosis who received three different biologic classes for disease treatment. We included all patients who received at least one dose of the third-class biologic. Only patients with active disease (defined as Harvey–Bradshaw index (HBI) ≥ 5) were included. Patients with ostomy or pouch surgery were excluded. Patients that had not yet reached the primary endpoint (week 16) while on active third-class therapy were also excluded.

The study was approved by the Sheba Medical Center ethics committee and written, informed consent was obtained from each patient included in the study. Approval was granted for Helsinki protocol SMC-5598-08 on 28 January 2009. The study protocol conforms to the ethical guidelines of the 1975 Declaration of Helsinki, as reflected in a priori approval by the institution’s human research committee.

### 2.2. Clinical Scores and Outcomes

The primary outcome of the study was clinical response at week 16–22 (defined by a reduction of HBI ≥ 3). Secondary outcomes included clinical remission (HBI ≤ 4) at week 16–22, clinical remission and response in week 52, steroid-free clinical remission at week 16–22, C-Reactive Protein (CRP) normalization (CRP serum concentration levels less than normal range as per the cut-off used in the corresponding institutions).

### 2.3. Statistical Methods

Continuous variables were articulated as the median and interquartile range (IQR). Categorical variables were analyzed using Chi-squared/Fisher’s exact tests and continuous variables by the *t*-test/Mann–Whitney test as appropriate. A *p*-value < 0.05 was considered statistically significant. We constructed a multivariate logistic regression model to identify the independent predictors of week 16–22 response and remission, as well as treatment continuation after induction. Variables with a significance level <0.1 based on univariate analysis were included in the multivariate model. To investigate the effect of the variables on treatment discontinuation, we planned a survival analysis using a Kaplan Meyer survival curve or Cox proportional hazard analysis as appropriate. The analysis was performed using IBM SPSS (version 22.0; Armonk, NY, USA).

## 3. Results

### 3.1. Baseline Characteristics

Two-hundred and four patients from 27 centers in 15 countries (22 Europe, 4 Israel, and 1 Canada) were included in the study. Baseline characteristics are detailed in Table 1. All patients had received anti-TNFs as a first-class biological therapy (51 patients had previously received one, 130 had received two, and 23 had received three anti-TNFs). One hundred and fifty-six out of 204 (76%) patients received VDZ as a second-class therapy (median treatment duration on VDZ was 12 months (IQR 5–20)) and UST as a third-class therapy (group A); the remaining 48 of 204 (24%) patients received UST as a second-class therapy (median treatment duration on UST was 10 months (IQR 6.5–18)) and VDZ as a third-class therapy (group B).

Median disease duration before the start of third-class treatment was 16 years (IQR 10–22). All patients had active CD (median HBI: 8 (IQR 7–12)) at the time of third-class onset.

One-hundred and thirty-two (64.7%) had elevated CRP at treatment onset. The patient characteristics were similar between the groups, with the exception of the prevalence of perianal disease (group A: 37.8% vs group B: 54.1%, *p* = 0.04). Figure 1 describes the patient flow during the study.

### 3.2. Treatment Outcomes

#### 3.2.1. Induction Period (Weeks 16–22)

The overall response rate at week 16–22 was 55.8% (87/156 (55.5%) and 27/48 (56.2%) in groups A and B, respectively (*p* = 0.9)). Clinical remission was achieved by 56/204 (27.4%) patients (41/156 (26.2%) and 15/48 (31.2%), (*p* = 0.5)) (Figure 2A). CRP normalized in 76/172 (44.18%) patients with CRP values available at this time point (59/134 (44.02%) and 17/38 (44.7%), (*p* = 0.9)) (Figure 2B). Systemic corticosteroids were discontinued in 51/80 (63.7%) patients who were on corticosteroids at the treatment onset. Corticosteroid-free remission was achieved by 18/80 (22.4%) (14/62 (22.5%) and 4/18 (22.2%), (*p* = 0.9)) (Figure 2B).

There was a significant negative association between elevated CRP and clinical response (*p* = 0.02). Clinical remission was negatively associated with elevated CRP and smoking (*p* = 0.03 and *p* = 0.01, respectively) (Table 2).

#### 3.2.2. Maintenance (Week 52)

One hundred and thirty-two patients had data available for analysis for week 52 (103/132 in group A and 29/132 in group B) (Figure 1). The overall response rate was 87.1% (89/103 (86.4%) and 25/29 (86.2%) in groups A and B, respectively (*p* = 0.9)). Clinical remission was achieved by 38/132 (28.7%) patients (31/103 (30%) and 7/29 (24.1%), (*p* = 0.5)) (Figure 2A). CRP normalized in 52/101 (51.4%) patients with CRP values available at this time point (42/80 (51.2%) and 10/21 (47.6%), (*p* = 0.6)) (Figure 2B). Systemic corticosteroids were discontinued in 34/45 (75.5%) patients who were on corticosteroids at the time of third-class agent initiation. Corticosteroid-free remission was achieved by 12/45 (26.6%) patients (11/36 (30.5%) and 1/9 (11.11%), (*p* = 0.2)) (Figure 2B).

Seventy-two patients had no data available for analysis in week 52 (53/156 in group A and 19/48 in group B). The last follow-up in these patients was with a median of 21 weeks (IQR 17–38). The overall response rate was 52.7% (28/53 (52.8%) and 10/19 (52.6%) in groups A and B, respectively (*p* = 0.4)). Clinical remission was achieved by 18/72 (25%) patients (14/53 (26.4%) and 4/19 (21%), (*p* = 0.3)). CRP normalized in 12/46 (26%) patients with CRP values available at this time point (10/33 (30.3%) and 2/13 (15.3%), (*p* = 0.1)). Systemic corticosteroids were discontinued in 17/30 (56.6%) patients who were on corticosteroids at the time of third-class agent initiation. Corticosteroid-free remission was achieved by 8/30 (26.6%) patients (7/23 (30.4%) and 1/7 (14.2%), (*p* = 0.2)).

Dose escalation was required in 104/204 (50.9%) patients (74/156 (47.4%) in group A with a median time of 26 weeks (IQR 8–52) and 30/48 (62.5%) in group B with a median time of 26 weeks (IQR 15–50), respectively (*p* = 0.03); sixty-nine out of them (66.3%) achieved clinical remission or response by the end of the follow-up (39/74 (52.7%) and 30/48 (62.5%), respectively, (*p* = 0.14)).

#### 3.2.3. Treatment Discontinuation

Seven patients stopped the treatment before week 16, with a median of 10 weeks (IQR 8–12), mainly due to a primary non-response. The overall treatment discontinuation was observed in 19.11% of the patients (36/204), with a median follow-up duration of 48 (IQR 21–52) weeks. Twenty-seven patients out of 156 (17.3%) in group A with a median time of 21 weeks (IQR 16–20) and 9/48 (18.75%) in group B with a median time of 36 weeks (20–45) discontinued treatment (hazard ratio, 1.01; 95% confidence interval (CI), 0.49–2, *p* = 0.9). The Kaplan–Meier curve for discontinuation-free survival is presented in Figure 3.

The main reason for the discontinuation of both treatment groups was a lack of response (group A: 85%, group B: 77.7%). The rest discontinued the treatment because of: patient decision (*n* = 2), adverse event (3), active extraintestinal manifestation (1).

On univariable analysis, treatment continuation was negatively associated with disease duration at third-class agent onset, and stricturing disease behavior (*p* = 0.008, (odds ratio (OR) 0.93; 95% CI 1–1.1), *p* = 0.004 (OR 2.1; 95% CI 0.1–1.1), respectively); and positively associated with immunomodulatory drug use (*p* = 0.001, OR 0.04; 95% CI 0.01–0.18). However, on multivariate analysis, only concomitant use of immunomodulatory drug use was positively associated (relative risk (RR) 0.014, 95% confidence interval (CI) 0.001–0.31) and disease duration at third-class agent onset was negatively associated with treatment continuation (Table 2).

#### 3.2.4. Safety

Adverse events during the follow-up occurred in 19/204 (9.31%) patients. Thirty patients out of the 204 (14.7%) were hospitalized; the reasons for hospitalization are detailed in Table 3. Twenty-seven patients out of the 204 (13.2%) required surgical intervention, as detailed in Table 3.

## 4. Discussion

Our study demonstrates that third-class biologic therapy is a feasible, effective, and safe option in Crohn’s disease. It is well established that the number of prior treatments is negatively associated with therapeutic success [29,31,32], and that biologic-naïve patients have a substantially higher likelihood of a response to treatment in inflammatory bowel diseases (IBDs) [17,33,34,35,36,37]. Until a few years ago, our therapeutic armamentarium in IBD was limited to TNF-alpha inhibitors only, but since then it has expanded to include anti-integrins, IL12/23 blockers, and JAK inhibitors. The natural history of utilization of any new drug naturally favors its use in patients who failed previous and better established treatments: thus, early real-world experience (RWE) series with VDZ included few biologic-naïve patients [18,21,26,38], whereas later series addressing biologic-naïve patients demonstrated higher effectiveness in comparison to those early publications [17,39,40]; for UST, data for biologic-naïve patients is still sparse. Similarly, the literature regarding the use of third-class biologics in IBD is limited to small numbers of patients in RWE series of biologic-experienced patients [27,41,42,43]. To date, the personalization of therapeutic decisions in IBD is underdeveloped, with limited clues to suggest which succession of therapeutic regimens is superior and subsequently likely to work best in patients failing several consecutive biologic treatments.

Our study focused on the effectiveness and safety of a third-class biologic in patients who had failed two previous classes s of biologics. The overall effectiveness of the third-class agent was quite comparable to previous RWE reports on UST and VDZ [15,44]. In several previous studies, UST appeared to be more effective than VDZ in CD patients refractory to anti-TNFs. However, these studies included UST or VDZ as second-class agents and not as third-class agents [27,28].

Adverse events were rare and in line with the previous reports as well. In our study, concomitant immunomodulatory therapy was not associated with the likelihood of response or remission, in similarity with the recently published data [45,46]; nonetheless, the risk of treatment discontinuation was higher in patients on monotherapy. The effect of concomitant immunomodulators in the present study could be explained by a decrease in the risk of anti-drug antibody formation in line with the recently reported data on sequential anti-TNF use [47] or due to an additional anti- inflammatory effect [48,49]. Another somewhat surprising finding was the lack of association of the number of previously used anti-TNFs with therapeutic outcomes. A possible explanation for this observation could stem from the fact that all our patients have already experienced failure of multiple biologics, potentially obviating the subtle differences that could have been observed in a more treatment-naïve cohort.

Despite the fact that 19.11% of the patients discontinued the therapy, the majority continued the therapy with a high response (87.1%) and remission rate (28.7%). Importantly, as the patients included in this cohort have already experienced multiple classes of biological therapy and have quite exhausted their treatment options, it is plausible that in a proportion of cases the treatment was continued for a longer duration of time despite the lack of a clear early therapeutic response.

Unfortunately, the groups (VDZ or UST as a third-class agent) were dissimilar in size; the likely explanation for this is the earlier introduction of VDZ into clinical practice. Although we could not demonstrate any significant difference between the groups in either clinical characteristics or outcomes, the comparison is compromised by the disproportional cohort size. Some of the additional limitations of our study are inherent to an RWE multicenter retrospective study design (including a small number of patients with available endoscopic data; missing laboratory data, including drug levels and anti-drug antibodies; and heterogeneity in the scheduled visit dates, follow up data about perianal disease, and extraintestinal manifestations). Therefore, no conclusion could be drawn on the effect of these third-class biologicals on endoscopic endpoints and perianal disease. The study cohort was narrow in terms of age (41.5 years old (IQR 32–53)) and may not well reflect the responses of all patients. There is no representation for the elderly nor the young under the age of 30.

Despite the aforementioned limitations, our study supports the use of third-class biologics in CD. Further research is required to identify the most effective succession of treatment regimens in IBD, and to further personalize decision-making on those challenging patients.

## Figures and Tables

**Figure 1 jcm-10-02914-f001:**
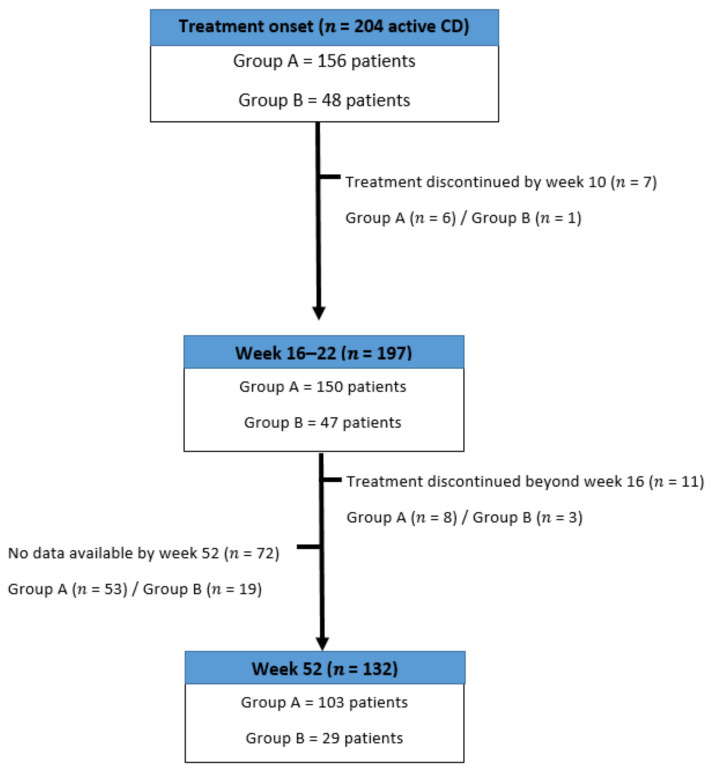
Chart of the patients’ flow during the study period. CD, Crohn’s disease; group A, patients received vedolizumab (VZD) as a second-class therapy and ustekinumab (UST) as a third-class therapy; group B, patients received UST as a second-class therapy and VDZ as a third-class therapy.

**Figure 2 jcm-10-02914-f002:**
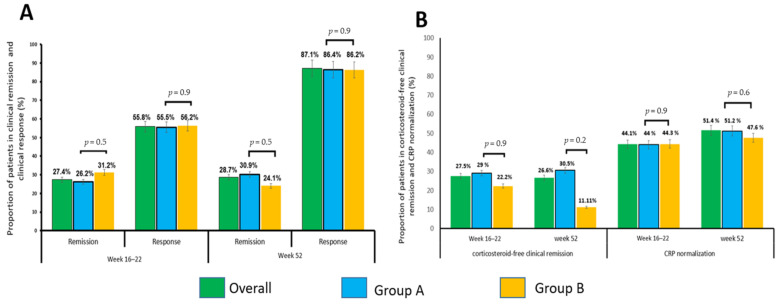
Outcome. (**A**): The proportion of patients achieving clinical remission (HBI ≤ 4) and clinical response (delta HBI ≥ 3) during the study period. (**B**): The proportion of patients achieving corticosteroid-free clinical remission during the study period, and the proportion of patients with CRP normalization (CRP serum concentration levels less than normal range as per cut-off used in the corresponding institutions). CRP, C-reactive protein, HBI: Harvey–Bradshaw index.

**Figure 3 jcm-10-02914-f003:**
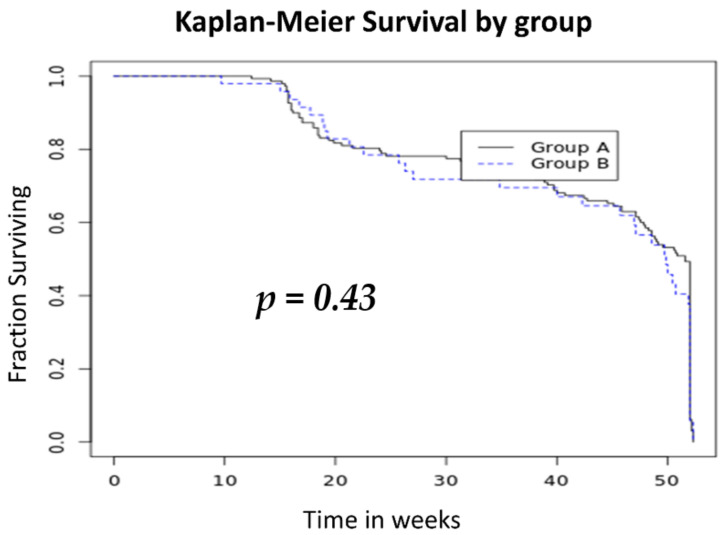
Kaplan–Meier curve of treatment discontinuation-free analysis. Abbreviations: Group A, patients receiving vedolizumab (VZD) as a second-class therapy and ustekinumab (UST) as a third-class therapy; Group B, patients receiving UST as a second-class therapy and VDZ as a third-class therapy.

**Table 1 jcm-10-02914-t001:** Patients’ demographic and clinical characteristics.

	Total	Group A	Group B	*p*-Value
*n* (%)	204	156 (76.4%)	48 (23.6%)	*p* < 0.01 *
Age at start of 3rd-class biological therapy, years, median (IQR)	41.5 (32–53)	43.6 (32–52.7)	43.5 (31–54.7)	*p* = 0.96
Gender—male, *n* (%)	96 (47.05%)	74 (47.4%)	22 (45.8%)	*p* = 0.84
Current smoker, *n* (%)	36 (17.6%)	27 (17.3%)	9 (18.7%)	*p* = 0.81
Age at diagnosis, years median (IQR)	23 (17–32)	24 (18–34)	20 (15–31.7)	*p* = 0.13
Disease duration, years. median (IQR)	16 (10.25–22)	15 (10–20.7)	16.5 (11–25.75)	*p* = 0.13
Location (*n*, %)				
Ileum	55 (26.9%)	42 (26.9%)	13(27%)	*p* = 0.98
Colon	41 (20.09%)	36(23. %)	5 (10.1%)	*p* = 0.06
Ileocolonic	107 (52.45%)	78 (50%)	29 (60.4%)	*p* = 0.20
UGI	1 (0.49%	0 (0%)	1 (2%)	*p* = 0.16
Behavior (*n*, %)				
Nonstricturing nonpenetrating	70 (43.31%)	55 (35.2%)	15 (31.2%)	*p* = 0.60
Stricturing	55 (26.96%)	46 (29.4%)	9 (18.7%)	*p* = 0.14
Penetrating	64 (31.37%)	48 (30.7%)	16 (33.3%)	*p* = 0.73
Peri-anal disease, *n* (%)	85 (41.66%)	59 (37.8%)	26(54.1%)	*p* = 0.04 *
Prior anti-TNF therapy exposure, *n* (%)				
One previous anti-TNF treatment	51 (25%)	42 (26.9%)	9(18.7%)	*p* = 0.25
Two previous anti-TNF treatments	130 (63.72%)	100 (64.1%)	30 (62.5%)	*p* = 0.84
Three previous anti-TNF treatments	23 (11.27)	14 (8.9%)	9 (18.7%)	*p* = 0.06
Disease activity				
Harvey Bradshaw Index, Median (IQR)	8 (7–12)	8 (7–11)	9 (6–12)	*p* = 0.24
Elevated CRP, *n* (%)	132 (64.7%)	101 (64.7%)	31 (64.5%)	*p* = 0.98
Fecal calprotectin > 250 µg/g, *n* (%)	73 (35.78%)	52 (33.3%)	16 (33.3%)	*p* = 1.00
Concomitant medications, *n* (%)				
Corticosteroids	80 (39.21%)	62 (39.7%)	18 (37.5%)	*p* = 0.78
Immunosuppressant	50 (24.5%)	37 (23.7%)	13 (27%)	*p* = 0.63
Both corticosteroids and immunosuppressant	18 (8.82%)	12 (7.69%)	6 (12%)	*p* = 0.30

Abbreviations: *n*, number; IQR, interquartile range; UGI, upper gastrointestinal; anti-TNF, anti-tumor necrosis factor; CRP, C-reactive protein; *, significant *p*-value; Group A = ustekinumab, Group B = vedolizumab.

**Table 2 jcm-10-02914-t002:** Clinical variables associated with clinical response, remission, and treatment discontinuation.

	*p*-Value for Clinical Response (Week 16–22)	*p*-Value for Clinical Remission (Week 16–22)	*p*-Value for Treatment Discontinuation		OR for Treatment Discontinuation
			Univariate	Multivariate	
Group	0.95	0.5	0.8	-	OR, 0.9; 95% CI, 0.4–2.5
Gender	0.33	0.63	0.08	-	OR, 1.5; 95% CI, 0.3–1.3
Age at start of 3rd-class biological therapy	0.97	0.74	0.6	-	OR, 0.98; 95% CI, 0.9–1
Age at diagnosis	0.94	0.73	0.1	-	OR, 1.11; 95% CI, 0.9–1
Disease duration at 3rd-class onset	0.74	0.4	0.01 *	<0.01 *	OR, 0.93; 95% CI, 1–1.1
Current smoker	0.84	0.01 *	0.09	-	OR, 1.8; 95% CI, 0.1–2
Location	0.9	0.11	0.6	-	OR, 0.8; 95% CI, 0.5–3
Behavior	0.25	0.34	<0.01 *	-	OR, 2.1; 95% CI, 0.1–1.1
Prior anti-TNF therapy exposure	0.08	0.06	0.6	-	OR, 1.2; 95% CI, 0.4–1.5
Elevated CRP	0.02 *	0.03 *	0.2	-	OR, 0.5; 95% CI, 0.7–5
Corticosteroids	0.23	0.34	0.2	-	OR, 1.7; 95% CI, 0.2–1.2
Concomitant immunomodulators	0.14	0.29	<0.01 *	0.04 *	OR 0.04; 95% CI, 0.01–0.18

Abbreviations: *n*, number; IQR, interquartile range; OR, odds ratio; UGI, upper gastrointestinal; anti-TNF, anti-tumor necrosis factor; CRP, C-reactive protein; *, significant *p*-value; Group A = ustekinumab, Group B = vedolizumab.

**Table 3 jcm-10-02914-t003:** Adverse events, hospitalization, and surgeries during the follow-up period in each group.

	Group A	Group B
Adverse event		
Arthralgia	4	2
Skin reaction d/t infusion	-	1
Recurrent infection	4	-
Psoriasis	1	1
Depression	2	-
Severe anemia	-	1
Dermatitis/eczema	-	1
Occlusion due to stenosis with surgery	1	-
New heart failure	1	-
Hospitalization		
Flare	10	10
SBO	2	-
Perianal abscess	1	1
Depression	1	1
Enterocutaneous fistula	1	1
Acute appendicitis	1	-
Diverticulitis	-	1
Surgery		
Small bowel resection	3 *	2
Perianal abscess	-	3
Rt. colectomy	1	-
Total colectomy	1	-
Ileocolonic resection	8	3
Fistulotomy	1	2
Rectal cancer resection	-	1
Colostomy	1	-
Rectal surgery	1	-

Abbreviations: SBO, small bowel obstruction; *, combined with fistulotomy.

## Data Availability

No new data were created or analyzed in this study. Data sharing is not applicable to this article.

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
