# Peer review of "Effectiveness of Third-Class Biologic Treatment in Crohn’s Disease: A Multi-Center Retrospective Cohort Study"

_jcm, 2021, doi:10.3390/jcm10132914_

Round 1
Reviewer 1 Report
I congratulate the authors on presenting a retrospective, multicentre cohort study of relatively treatment resistant (failed 2x previous biologics) Crohn’s patients comparing third-line biologic therapy with UST (with previous anti-TNF and VDZ) to VDZ (with prior anti-TNF and UST). Response/remission and treatment persistence rates were similar between the groups. The two groups were relatively similar, apart from a higher rate of perianal disease among the VDZ third-line group. However, the study is significantly limited, as the authors acknowledge, by the relatively small UST group, likely as a result of the more recent introduction of UST into clinical practice. Despite this significant limitation, the data has clinical value. It could be strengthened by addressing the following points:
Major
- It is concerning that data was not available for such a large proportion of patients at week 52 (53/150 group A and 19/47 group B), weakening the strength of conclusions for the response/remission data at week 52. Can you present some week 16 response or CRP data on the group of people without week 52 data – ie, were they responders or non-responders at week 16-22? What was their CRP / calpro data at week 16?
- It would be helpful to know the proportions of patients in each group who required escalation of their biologics, whether this was associated with improved response rates, and whether there was a difference in escalation rates between the groups - do you have any data on biologic escalation of the VDZ or UST groups?
- Figure 1 is somewhat confusing with regard to the direction of the arrowheads. It could be improved if the arrows of the flow chart faced in the direction of the flow of the patients (ie. with patients who discontinued, the arrowheads should point towards the discontinuation box).
- Table 2 should have some sort of quantifying statistics, ie odds ratio
- Similar to above, in section 3.2.3, when discussing the univariable analysis, please include ORs.
- In the discussion, the line about “recently reported data on sequential TNF use” needs a reference. However, I don’t think the recently reported data on sequential anti-TNF use showing use of concomitant immunomodulators reducing immunogenicity to the second TNF in patients who had immunogenicity to the first TNF (Roblin, Gut 2020) is entirely applicable here, given that anti-TNFs are known to be significantly more immunogenic than VDZ/UST which have low immunogenicity. In this relatively treatment refractory cohort, could the benefit seen with immunomodulators be a therapeutic effect of the immunomodulator on the disease itself?
- Regarding time to treatment discontinuation, while there was a strong treatment persistence rate with similarity between groups, can it not also be explained by persistence despite treatment ineffectiveness in the face of a lack of alternative therapies? I think this should be commented upon.
Minor:
- Line 103 – “The primary outcome of the study was clinical response (defined by a reduction of 103 HBI ≥ 3)” – I am assuming this is at week 16-22 – could you please confirm this?
- Line 182 – is this supposed to be a Hazard Ratio as opposed to an odds ratio?
- Table 3 – correct the spelling of “endocutaneous”
- Table 3 – I’m presuming the use of the word “short” throughout the table and the abbreviations should be “small”
- Line 246 – change “date” to “data”
Author Response
Dear Editor
We would like to thank the reviewers for their time and insightful comments. We enclose a revised version of the manuscript.
Regarding specific comments by reviewers:
Referee 1 comments:
I congratulate the authors on presenting a retrospective, multicentre cohort study of relatively treatment resistant (failed 2x previous biologics) Crohn’s patients comparing third-line biologic therapy with UST (with previous anti-TNF and VDZ) to VDZ (with prior anti-TNF and UST). Response/remission and treatment persistence rates were similar between the groups. The two groups were relatively similar, apart from a higher rate of perianal disease among the VDZ third-line group. However, the study is significantly limited, as the authors acknowledge, by the relatively small UST group, likely as a result of the more recent introduction of UST into clinical practice. Despite this significant limitation, the data has clinical value. It could be strengthened by addressing the following points:
Major
- It is concerning that data was not available for such a large proportion of patients at week 52 (53/150 group A and 19/47 group B), weakening the strength of conclusions for the response/remission data at week 52. Can you present some week 16 response or CRP data on the group of people without week 52 data – ie, were they responders or non-responders at week 16-22? What was their CRP / calpro data at week 16?
R- Thank you for the comment. The response rate at week 16 was 55.8%, remission rate 27.4% and CRP-normalization rate 44.1%. The data now appears in paragraph 3.2.1 as suggested by the reviewer
- It would be helpful to know the proportions of patients in each group who required escalation of their biologics, whether this was associated with improved response rates, and whether there was a difference in escalation rates between the groups - do you have any data on biologic escalation of the VDZ or UST groups?
R- Dose escalation was required in 50.9% of the patients (47.4% of the patients in group A and 62.5% of the patients in group B). The data now appears in section 3.2.2 as suggested by the reviewer
- Figure 1 is somewhat confusing with regard to the direction of the arrowheads. It could be improved if the arrows of the flow chart faced in the direction of the flow of the patients (ie. with patients who discontinued, the arrowheads should point towards the discontinuation box).
R- This was amended as requested
- Table 2 should have some sort of quantifying statistics, ie odds ratio
R- This was amended as requested
- Similar to above, in section 3.2.3, when discussing the univariable analysis, please include ORs.
R- Done
- In the discussion, the line about “recently reported data on sequential TNF use” needs a reference. However, I don’t think the recently reported data on sequential anti-TNF use showing use of concomitant immunomodulators reducing immunogenicity to the second TNF in patients who had immunogenicity to the first TNF (Roblin, Gut 2020) is entirely applicable here, given that anti-TNFs are known to be significantly more immunogenic than VDZ/UST which have low immunogenicity. In this relatively treatment refractory cohort, could the benefit seen with immunomodulators be a therapeutic effect of the immunomodulator on the disease itself?
R- We added this reservation in the discussion as suggested by the reviewer
- Regarding time to treatment discontinuation, while there was a strong treatment persistence rate with similarity between groups, can it not also be explained by persistence despite treatment ineffectiveness in the face of a lack of alternative therapies? I think this should be commented upon.
R- We agree with the reviewers comment and added an appropriate comment to the discussion
Minor:
- Line 103 – “The primary outcome of the study was clinical response (defined by a reduction of 103 HBI ≥ 3)” – I am assuming this is at week 16-22 – could you please confirm this?
R- This was confirmed in the manuscript.
- Line 182 – is this supposed to be a Hazard Ratio as opposed to an odds ratio?
R- Corrected in the manuscript.
- Table 3 – correct the spelling of “endocutaneous”
R- Corrected in the manuscript.
- Table 3 – I’m presuming the use of the word “short” throughout the table and the abbreviations should be “small”
R- Corrected in the manuscript.
- Line 246 – change “date” to “data”
R- Corrected in the manuscript. Thank you
We would like to thank the reviewers and the editorial team for the insightful comments
We hope that the revised version will merit publication in JCM
Sincerely,
Dr. Ahmad Albshesh
Reviewer 2 Report
In this well written retrospective multicenter study evaluating the effectiveness of third line biologic therapies for Crohn's disease the authors conclude that third line therapy is effective.
Methods : it seems the authors performed unpaired t-tests when comparing effectiveness at different timepoints ( HBI before and after , CRP before and after etc..) see results comments bellow - if paired analyses are performed this should stated as well
Results: The authors give information regarding response rates with p values after treatment ( at weeks 16-22 and 52 ) for the different endpoints - and state for example that 132 patients had elevated CRP before treatment onset and CRP normalized in 44% of patients (76/172) . It seems these are pooled data ( CRP available before and after with analyses by unpaired t-tests ) . It would be informative to know how many patients with elevated HBI, CRP, FCP etc at baseline went on to normalize these respective endpoints at weeks 16 and 52 ( ie patient data available at both baseline and weeks 16 and 52 ) . If these data are available paired t-test should then be performed in order to evaluate significance
Author Response
Dear Editor
We would like to thank the reviewers for their time and insightful comments. We enclose a revised version of the manuscript.
Regarding specific comments by reviewers:
Referee 2 comments:
In this well written retrospective multicenter study evaluating the effectiveness of third line biologic therapies for Crohn's disease the authors conclude that third line therapy is effective.
- Methods : it seems the authors performed unpaired t-tests when comparing effectiveness at different timepoints ( HBI before and after , CRP before and after etc..) see results comments bellow - if paired analyses are performed this should stated as well
R- This was unpaired t-test- see further comment in the results comments.
- Results: The authors give information regarding response rates with p values after treatment ( at weeks 16-22 and 52 ) for the different endpoints - and state for example hat 132 patients had elevated CRP before treatment onset and CRP normalized in 44% of patients (76/172) . It seems these are pooled data ( CRP available before and after with analyses by unpaired t-tests ) . It would be informative to know how many patients with elevated HBI, CRP, FCP etc at baseline went on to normalize these respective endpoints at weeks 16 and 52 ( ie patient data available at both baseline and weeks 16 and 52 ) . If these data are available paired t-test should then be performed in order to evaluate significance
R- Unfortunately, the CRP value are from different centers with different cutoffs. Thus we used the values as a binary variable (normal/abnormal) and not as continuous variables, making the use of paired parametric tests impossible.
We would like to thank the reviewers and the editorial team for the insightful comments.
We hope that the revised version will merit publication in JCM
Sincerely,
Dr. Ahmad Albshesh
Round 2
Reviewer 1 Report
Thankyou for addressing the majority of my comments.
Just a few things to address further:
- It would still be helpful to present week 16-22 data on the large group of patients (53/150 group A and 19/47 group B) who continued their drug but for whom week 52 data was not available, in order to make sure response rates were not too different from the overall group. Can you present this group’s response and CRP data at week 16-22? This may fit into section 3.2.2
- In table 2 – it would be useful to show quantifying statistics (ie. odds ratios and confidence intervals) for all p values presented, or at the very least all significant p values, in order to help with interpretation of these results.
- In section 3.2.3, could you include an OR for disease duration at third-class agent onset to support the sentence “treatment continuation was negatively associated with disease duration at third-class agent onset”. Furthermore, while it is written that “treatment continuation was negatively associated with… disease behavior”, disease behaviour seemed to be associated with a lower odds of treatment discontinuation (odds ratio (OR), 0.24; 95% CI, 0.11-0.50) as per table 2. Could you clarify this, also clarifying how you dealt with disease behavior as a variable (ie was it dichotomized) in order to aid in the clinical interpretation of this sentence?
Author Response
Dear Editor,
We would like to thank the reviewers again for their time and insightful comments. Regarding specific comments by reviewers:
1. It would still be helpful to present week 16-22 data on the large group of patients (53/150 group A and 19/47 group B) who continued their drug but for whom week 52 data was not available, in order to make sure response rates were not too different from the overall group. Can you present this group’s response and CRP data at week 16-22? This may fit into section 3.2.2
R- The overall response/remission rate in patients without available data in week 52 was 52,7% and 25% respectively, CRP normalization was achieved in 26%. The data now appears in section 3.2.2 as suggested by the reviewer
2. In table 2 – it would be useful to show quantifying statistics (ie. odds ratios and confidence intervals) for all p values presented, or at the very least all significant p values, in order to help with interpretation of these results.
R- The OR appears in table 2 as requested by the reviewer.
3.In section 3.2.3, could you include an OR for disease duration at third-class agent onset to support the sentence “treatment continuation was negatively associated with disease duration at third-class agent onset”, furthermore, while it is written that “treatment continuation was negatively associated with… disease behavior”, disease behavior seemed to be associated with a lower odds of treatment discontinuation (odds ratio (OR), 0.24; 95% CI, 0.11-0.50) as per table 2. Could you clarify this, also clarifying how you dealt with disease behavior as a variable (ie was it dichotomized) in order to aid in the clinical interpretation of this sentence?
R- Thank you for noticing this error, we have recalculated the OR and rephrased the sentence.
We would like to thank the reviewers and the editorial team and we hope that the revised version will merit publication.
Sincerely,
Dr. Ahmad Albshesh
This manuscript is a resubmission of an earlier submission. The following is a list of the peer review reports and author responses from that submission.